# Knowledge, attitude, practice, and fear level of Bangladeshi students toward Covid-19 after a year of the pandemic situation: A web-based cross-sectional study

**Tahsin Ahmed Rupok**[ID][1]*, **Sunandan Dey**[1], **Rashni Agarwala**[2], **Md. Nurnobi Islam**[3], **Bayezid Bostami**[1]

**1** Department of Pharmacy, University of Rajshahi, Rajshahi, Bangladesh, **2** Department of Pharmacy, Islamic University, Kushtia, Bangladesh, **3** Department of Chemistry, Shahjalal University of Science & Technology, Sylhet, Bangladesh

* tahsinahmedrupokpharmru9176@gmail.com

**Data Availability Statement:** All relevant data are within the manuscript and its supporting information files.

## Abstract

### Introduction

In the earlier phase of the pandemic situation, the Government of Bangladesh (GoB) badly suffered to adhere their people to preventive measures probably due to less knowledge and attitude toward Covid-19. To tackle the second wave of coronavirus, the GoB has again enforced an array of preventive measures, but still encountering the same problem after a year of the pandemic situation. As an attempt to find out the reasons behind this, our study aimed to assess the present knowledge and fear level regarding Covid-19, and attitude and practice of students toward Covid-19 preventive measures (CPM).

### Methods

A cross-sectional study was designed and conducted from 15th to 25th April 2021. A total of 382 participants met all the inclusion criteria and were considered for performing all the statistical analyses (Descriptive statistics, Mann-Whitney U test, Kruskal-Wallis H test, Multiple logistic regression, Spearman rank-order correlation).

### Results

All the participants were students aged 16 to 30 years. 84.8%, and 22.3% of participants had respectively more accurate knowledge, and moderate to high fear level regarding Covid-19. And, 66%, and 55% of participants had more positive attitude, and more frequent practice toward CPM, respectively. Knowledge, attitude, practice, and fear were interrelated directly or indirectly. It was found knowledgeable participants were more likely to have more positive attitude (AOR = 2.34, 95% CI = 1.23–4.47, P < 0.01) and very little fear (AOR = 2.17, 95% CI = 1.10–4.26, P < 0.05). More positive attitude was found as a good predictor of more frequent practice (AOR = 4.00, 95% CI = 2.44–6.56, P < 0.001), and very less fear had negative impact on both attitude (AOR = 0.44, 95% CI = 0.23–0.84, P < 0.01) and practice (AOR = 0.47, 95% CI = 0.26–0.84, P < 0.01).

**Funding:** The authors received no specific funding for this work.

**Competing interests:** The authors have declared that no competing interests exist.

## Conclusions

The findings reflect that students had appreciable knowledge and very little fear, but disappointedly had average attitude and practice toward Covid-19 prevention. In addition, students lacked confidence that Bangladesh would win the battle against Covid-19. Thus, based on our study findings we recommend that policymakers should be more focused to scale up students' confidence and attitude toward CPM by developing and implementing well-conceived plan of actions besides insisting them to practice CPM.

## Introduction

The Covid-19 pandemic had a devastating impact on both human life and national economy of Bangladesh, a middle-income country of South Asian region. Though the crisis has been precipitated in recent months, there is a potentiality of re-emergence of the pandemic again. If the pandemic starts to escalate vehemently in Bangladesh as it did before, the number of cases could rise to an uncontrollable digit, as majority of the population are lackadaisical in maintaining public health hygiene which makes the government's preventive plans inconvenient to implement [1].

It is worth mentioning that prevention is the best and comparatively less costly method to mitigate the pandemic. Only adequate preventive measures can keep the infection rate under control, which will truncate both the number of patients and death tallies. During the initial phases of the pandemic, government implemented generalized lockdown [2]. However, lockdowns are only effective if they are accompanied by other preventive measures [3]. The government of Bangladesh badly suffered to adhere their population to preventive measures in the earlier phase of the pandemic situation probably due to people's less knowledge and attitude to embrace the measures taken by the government. A review of some previous studies conducted in Bangladesh reflected the truth that Bangladeshi people were comparatively less knowledgeable, having a less positive attitude toward Covid-19, and as a consequence having less frequent preventive practice [4–8] (Fig 1).

However, one thing to remember, adhering people to preventive measures is not straightforward, takes intensive efforts over a length of time as well as depends on what extent people can perceive the importance of those measures. So, proper implementation of these measures

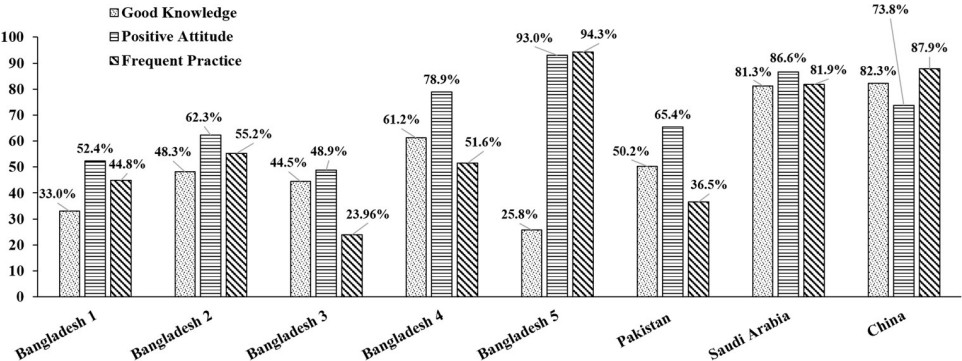

**Fig 1. Diagram reflecting different countries and their peoples' knowledge, attitude, and practice level toward covid-19.** *[Bangladesh 1* [4]*, Bangladesh 2* [6]*, Bangladesh 3* [8]*, Bangladesh 4* [7]*, Bangladesh 5* [5]*, Pakistan* [10]*, Saudi Arabia* [11]*, China* [12]*].*

warrants adequate well-planned initiatives to scale up people's knowledge and awareness toward Covid-19 and its preventive measures, which will ultimately alter their behavior as well as attitude positively toward Covid-19 preventive practice [8, 9]. Moreover, people should be resilient to this new situation, learn all fundamental information regarding Covid-19, and change their attitude positively toward practicing all safety guidelines stipulated by their government.

After a year of pandemic, to what extent people of Bangladesh have adapted to this new situation, changing their attitude and adhering themselves to Covid-19 preventive practice are needed to explore, which has become paramount because the government is still confronting the same troubles what did they face in the early phase of pandemic in adhering their people to Covid-19 preventive measures (CPM). Therefore, there should have a study to explore present Knowledge, Attitude, and Practice (KAP) level of Bangladeshi people at the end of a year of pandemic. However, our study has been designed to bring up the present KAP level of Bangladeshi students after a year of pandemic and how well-prepared they are to confront any future onslaught of Covid-19. So, the objectives of this present study were-

- To explore knowledge and fear level of Bangladeshi students regarding Covid-19 and their associated factors.

- To explore attitude and practice of students toward CPM and their associated factors.

- To find out whether knowledge, attitude, practice, and fear are related to each other.

- To find out the most probable reasons why students are digressed from maintaining CPM enforced by the government.

## Materials and methods

### Participants and data recruitment procedure

This web-based cross-sectional survey was conducted on Bangladeshi students amid the second wave of coronavirus in Bangladesh, continued from 15th to 25th April in 2021. A structured questionnaire was prepared in a Google Form which was shared with all authors and other volunteers who were employed for recruiting data from participants. A face-to-face community-based survey was not possible as the government implemented their second strict lockdown from 14th April being one day ahead of our data collection initiated [13]. All data were therefore assembled through online approaches using Whatsapp, Messenger, Email, Facebook. This survey included subjects aged 16 years old or above, being Bangladeshi residents, having internet access and ability to communicate, having no intellectual disability. Subjects living in foreign countries, having no or less interest to participate in were excluded from the survey. A total of 382 participants successfully submitted their data and all completed the questionnaire adequately as they were recommended.

### Survey instrument

The questionnaire prepared after a thorough review of existing literatures, contains following six sections.

**Preface.** This section had a brief delineation of the objectives of the survey and some instructions on how to complete the questionnaire properly. It also included a question asked to take consent of the respondents.

**Socio-demographic measures.** This section embodied 7 questions asked to gather socio-demographic information of a participant. Socio-demographic information included sex, age,

education, occupation, location, family type, and family income which was categorized as follows, lower income (<20,000 BDT), middle income (20,000–40,000 BDT), and higher income (>40,000 BDT).

**Knowledge scale.** This section had 12 questions where each question had three options 'Right', 'Wrong', and 'Not sure'. During scoring, responses of the questions with 'Wrong' as a correct answer (Table 2) were first recoded from 'Wrong' to 'Right'. Then, the correct answer (Right) was coded as 1 and the incorrect answer (Wrong/Not sure) was coded as 0. The total score ranged from 0 to 12 where those who obtained 83% score or greater ($\geq$ 10) were considered to have more accurate knowledge otherwise considered having less accurate knowledge. Instead of considering 80% score for more accurate knowledge based on Bloom's cut off point, 83% score was considered to avoid fractional digits as all the knowledge scores were integer numbers.

**Attitude scale.** This section had 8 questions where each question had three options 'Yes', 'No', and 'Maybe' except one question having 'Not possible' instead of 'Maybe'. (Table 2). During scoring, four kinds of options were coded as follows, 0 (No/ Not possible for work), 1 (Maybe), and 2 (Yes). The total score ranged from 0 to 16. Those who had an 81% score or greater ($\geq$ 13) were considered to have a positive attitude otherwise considered to have a less positive attitude. Instead of considering 80% score for positive attitude based on Bloom's cut off point, 81% score was considered to avoid fractional digits as all the attitude scores were integer numbers.

**Practice scale.** This section also had 8 questions where each question had three options 'Yes', 'No', and 'Sometimes'. During scoring, three options were coded as follows, 0 (No), 1 (Sometimes), and 2 (Yes). The total score ranged from 0 to 16. Those who had an 81% score or greater ($\geq$ 13) were considered to have more frequent practice otherwise considered having less frequent practice. Instead of considering 80% score for more frequent practice based on Bloom's cut off point, 81% score was considered to avoid fractional digits as all the practice scores were integer numbers.

**Fear scale.** This section included a fear measuring tool which was reported as a valid and reliable instrument to assess Covid-19 related fear among respondents [14]. This tool had 7 items where each item had five Likert options which were coded as follows, 0 (Strongly disagree), 1 (Disagree), 2 (Neutral), 3 (Agree), 4 (Strongly Agree). The total score ranged from 0 to 28. Fear scores greater or equal to 21 were cut-points for high fear. Scores between 20–14 were cut points for moderate fear and scores equal to or less than 13 were cut points for less fear.

This section also had two extra questions. The first question having binary options 'Yes', 'No' was asked if they are properly maintaining Covid-19 preventative measures stipulated by the government. The second question was asked to those who answered 'No' to account for their reasons why they are aberrant than those who answered 'Yes'. There were 17 possible responses and multiple responses could be selected.

**Reliability & validity.** Reliability tests of the instrument were performed using field data to determine validity and reliability of those scales mentioned above. Cronbach's alpha for knowledge, attitude, practice, and fear scale were 0.47, 0.61, 0.81, and 0.83. Due to a shortage of time, we could not conduct a pilot study.

## Ethical considerations

This study was conducted according to the principles expressed in the declaration of Helsinki. Besides, this study received ethical approval from Institute of Biological Science, University of Rajshahi, Bangladesh (ERC Memo no: 91/320IAMEBBC/IBSc). Anonymity and

confidentiality were strictly maintained. All respondents gave full informed consent to participate and consent for their data to be used in the publication. Participants aged under 18 years old were requested to provide their guardians' Email id or Messenger id or Whatsapp id or Phone number so that authors or volunteers can contact their guardians to inform them about the objectives of the survey, and take consents from them regarding their children's participation. This study included only those participants aged under 18 years old who made their guardians contact with authors or volunteers.

## Data management and analysis

All submitted data were automatically stored in a dynamic Microsoft excel sheet which was made available offline after completion of data collection for primary data processing including data duplication-checking, data cleaning, data coding, etc. In this stage, three data fallen out of inclusion criteria of age (below 16 years old), and two data got duplicated were discard from the preliminary dataset. After accomplishment of data cleaning and coding, the final dataset having data of 382 respondents was entered into Statistical Package for the Social Science (SPSS) version 26.0 for further data processing and doing all the statistical analysis which included descriptive statistics (to determine percentage, mean, standard deviation), Mann-Whitney U test (to determine difference between two groups under the same variable having two groups), Kruskal Wallis H test (to determine difference among the groups under same variable having more than two groups), Spearman rank-order correlation (if variables are ordinal), Multiple logistic regression (to determine predictors for accurate knowledge, positive attitude, frequent practice, and very little fear) including sex, education, occupation, family type, family income, location, knowledge, attitude, practice, fear as independent variables. Non-parametric Mann-Whitney U test and Kruskal Wallis H test, and Spearman rank-order correlation were performed as the data under continuous dependent variables were not normally distributed. To check that, besides visual inspection of histograms, Shapiro-Wilk and Kolmogorov-Smirnov tests were conducted. Both tests were statistically significant at 0.01 for those continuous dependent variables and histograms of them looked left-skewed, which confirmed well that data under those variables were not normally distributed. For all statistical analyses noted above, the statistical significance level was set to 0.05 (Alpha value).

## Results

### Demographic characteristics of the sample

This study considered a total of 382 participants for final statistical analysis. Among them majority were male (74.6%), studying in BSc and MSc (71.8%). All the participants were students while 39.8% were self-employed. About 76.7% and 58.1% of participants were from nuclear and lower income family (<20,000 BDT) respectively. 37.2% of participants informed their location 'village'. This study inclined toward the adult population as all participants had an age between 16 to 30 years. Table 1 tabulated detailed information about the socio-demographic characteristics of the respondents.

### Knowledge and its factors

Participants' overall correct answer rate was 89.6% [Mean Score (MS) ± Standard Deviation (SD) = 10.76±1.20 out of 12] indicating participants had appreciable knowledge on Covid-19. 84.8% of participants had more accurate knowledge while only 15.2% had less accurate knowledge. The distribution of responses for each question of the knowledge scale was presented in Table 2. Mann-Whitney and Kruskal-Wallis tests showed that knowledge scores do not vary

**Table 1. Socio-demographic characteristics of the respondents.**

| Characteristics | | Frequency | Percentage (%) |
|---|---|---|---|
| Sex | Male | 285 | 74.6 |
| | Female | 97 | 25.4 |
| Education | HSC | 104 | 27.2 |
| | BSc/MSc/Above | 278 | 71.8 |
| Occupation | Student and self-employed | 152 | 39.8 |
| | Student only | 230 | 60.2 |
| Family type | Nuclear | 293 | 76.7 |
| | Extended | 89 | 23.3 |
| Income | <20,000 BDT | 222 | 58.1 |
| | 20,000–40,000 BDT | 126 | 33.0 |
| | >40,000 BDT | 34 | 8.9 |
| Location | Village | 142 | 37.2 |
| | Sub-district town | 39 | 10.2 |
| | District town | 97 | 25.4 |
| | Divisional district town | 104 | 27.2 |

HSC- Higher Secondary School Certificate, BSc- Bachelor of Science, MSc- Masters of Science

with any socio-demographic variables included in our study (Table 4). Multiple logistic regression analysis identified location, attitude, and fear as significant predictors of more accurate knowledge on Covid-19 (Table 5).

## Attitude and its factors

On average, participants had an 82% score on the attitude scale (MS ± SD = 13.14 ± 2.21 out of 16). Approximately 66% of participants had more positive attitude. Only 35.3% of participants had confidence that Bangladesh could win the battle against Covid-19 and only 44.2% agreed that Covid-19 would finally be controlled. The distribution of responses for each question of the attitude scale was presented in Table 2. Mann-Whitney and Kruskal-Wallis tests showed attitude scores vary with occupation (P = 0.047) and family income (P = 0.05). It was found that the participants who were self-employed, and came from a lower income family (<20,000 BDT) had significantly less positive attitude than their counterparts (Table 4). Multiple logistic regression analysis identified income, location, knowledge, practice, and fear as significant predictors of more positive attitude. It was found that the participants who came of a lower income family (<20,000 BDT) were 67% less likely to have more positive attitude than those who came of a higher income family (> 40,000 BDT) (AOR = 0.33, 95% CI = 0.11–0.94, P < 0.05), and the participants who came from divisional district town were 50% less likely to have more positive attitude than those who came from village (AOR = 0.50, 95% CI = 0.27–0.93, P < 0.05) (Table 5).

## Practice and its factors

On average, participants had a 74.2% score on the practice scale (MS ± SD = 11.87 ± 3.80 out of 16). Approximately 55% of the participants had more frequent practice. The distribution of responses for each question of the practice scale was presented in Table 2. Mann-Whitney and Kruskal-Wallis tests uncovered that Practice scores vary with sex (P < 0.001), income (P < 0.001), and location (P < 0.01). It was found that females had significantly higher practice score than males (P < 0.001). Participants who came of both higher income family (P < 0.01)

**Table 2. Participants' knowledge, attitude, and practice towards covid-19 (N = 382) (distribution of responses).**

| Knowledge Questions (Correct Answer) | Right | Wrong | Not Sure |
|---|---|---|---|
| | n (%) | n (%) | n (%) |
| Covid-19 disease is deadly but preventable as well as curable. (Right) | 363 (95) | 7 (1.8) | 12 (3.1) |
| Fever, fatigue, dry cough, and breathing difficulty are the main clinical symptoms of Covid-19. (Right) | 357 (93.5) | 11 (2.9) | 14 (3.7) |
| Elderly people having chronic illnesses and other complications are more likely to be seriously affected. (Right) | 370 (96.5) | 2 (0.5) | 10 (2.5) |
| This disease can be prevented with antibiotics and other medications. (Wrong) | 89 (23.3) | 189 (49.5) | 104 (27.2) |
| A person with Covid-19 having no symptoms (fever) cannot spread the virus to others. (Wrong) | 30 (7.9) | 314 (82.2) | 38 (9.9) |
| Coronavirus can spread via respiratory droplets (from coughing, sneezing) of infected people. (Right) | 370 (96.9) | 8 (2.1) | 4 (1.0) |
| Taking measures is not necessary for children and young adults to prevent infections by the coronavirus. (Wrong) | 7 (1.8) | 360 (94.2) | 15 (3.9) |
| Isolation and treatment of people with Covid-19 are effective ways to reduce the spread of the virus. (Right) | 354 (92.7) | 14 (3.7) | 14 (3.7) |
| People who have contact with someone infected with the coronavirus or came from an infected area/country should be immediately isolated for 14 days. (Right) | 374 (97.9) | 4 (1.0) | 4 (1.0) |
| Covering the mouth and nose with a bent elbow or tissue paper or handkerchief when coughing or sneezing can reduce the spread of this virus. (Right) | 357 (93.5) | 16 (4.2) | 9 (2.4) |
| This disease could be asymptomatic. (Right) | 328 (85.9) | 23 (6.0) | 31 (8.1) |
| Individuals should avoid going to crowded places such as markets and public transportations to prevent infection. (Right) | 374 (97.9) | 6 (1.6) | 2 (0.5) |
| Attitude Questions | Yes | No | Maybe/Not Possible for work |
| | n (%) | n (%) | n (%) |
| Do you like to stay at home for a certain period (2 weeks) to prevent coronavirus spread if the government will order so? ** | 330 (86.4) | 24 (6.3) | 28 (7.3) |
| Do you think that social distancing (e.g. stay 1–2 m apart, avoid crowds, etc.) can prevent the spread of this virus? | 320 (83.8) | 14 (3.7) | 48 (12.6) |
| Do you agree that we should cancel business/ recreational trips at this time? | 249 (65.2) | 84 (22.0) | 49 (12.8) |
| Do you believe that working from home can help to control Covid-19? | 284 (74.3) | 40 (10.5) | 58 (15.2) |
| Do you agree that government should have taken preventive measures when new variants were first reported in Bangladesh? | 364 (95.3) | 3 (0.8) | 15 (3.9) |
| Do you think health education can play an important role in Covid-19 prevention? | 357 (93.5) | 7 (1.8) | 18 (4.7) |
| Do you agree that Covid-19 will finally be successfully controlled? | 169 (44.2) | 28 (7.3) | 185 (48.4) |
| Do you have confidence that Bangladesh can win the battle against the Covid-19 disease? | 135 (35.3) | 52 (13.6) | 195 (51.0) |
| Practice Questions | Yes | No | Sometimes |
| | n (%) | n (%) | n (%) |
| Do you wash hands with water and soap frequently? | 241 (63.1) | 30 (7.9) | 111 (29.1) |
| Do you always use a mask? | 305 (79.8) | 20 (5.2) | 57 (14.9) |

*(Continued)*

**Table 2.** (Continued)

| Knowledge Questions (Correct Answer) | Right | Wrong | Not Sure |
|---|---|---|---|
| Do you maintain the rules of using a mask? | 304 (79.6) | 23 (6.0) | 55 (14.4) |
| Do you maintain social distance (or home quarantine)? | 229 (59.9) | 66 (17.3) | 87 (22.8) |
| Do you use tissues or hanker chips during coughing/sneezing? | 228 (59.7) | 89 (23.3) | 65 (17.0) |
| Do you eat healthy food or maintain a healthy lifestyle focusing on outbreaks? | 255 (66.8) | 40 (10.5) | 87 (22.8) |
| Do you avoid public transports (like buses, trains, planes, etc.)? | 169 (44.2) | 123 (32.2) | 90 (23.6) |
| Do you avoid handshaking, hugging when you meet with your friends? | 223 (58.4) | 83 (21.7) | 76 (19.9) |

** This question had "Not possible for work" instead of "Maybe" in its three options. N- Number of respondents, %- Percentage of respondents, N- Total number of respondents

and middle income family (P < 0.01) had significantly higher practice score than those who came of a lower income family. Participants currently living in any district town and sub-district town (P ≤ 0.01) had significantly higher score on practice scale than those living in a village (Table 4). Multiple logistic regression analysis identified sex, location, attitude, and fear as significant predictors of more frequent practice. It was found that males were less likely to have more frequent practice than females by 49% (AOR = 0.51, 95% CI = 0.29–0.89, P ≤ 0.05). The participants living in any district town (AOR = 2.01, 95% CI = 1.09–3.65, P ≤ 0.05) or any divisional district town (AOR = 2.04, 95% CI = 1.23–4.47, P ≤ 0.01) were 2 times more likely to have more frequent practice than those living in a village. (Table 5).

## Fear and its factors

On average, participants had a 35.8% fear score (MS ± SD = 10.03 ± 4.77 out of 28). Of all, 77.7% (297) participants had less fear, 20.2% (77) had moderate fear, and only 2.1% (8) participants had high fear. Moreover, 77.7% (297) participants had very less fear (less than 50% fear score) while only 22.3% (85) participants had moderate to high fear (equal or more than 50% fear score). The distribution of responses of all fear-related questions was presented in Table 3. Mann-Whitney and Kruskal-Wallis tests unfolded that fear score among participants varies with only sex (P < 0.001). It was reported that females had a significantly high fear score than males (Table 4). Multiple logistic regression analysis detected sex, knowledge, attitude, and practice as significant factors associated with very little fear. It was found that male participants were 89% more likely to have very less fear than female participants (AOR = 1.89, 95% CI = 1.05–3.40, P < 0.05) (Table 5).

## Relationship among knowledge, attitude, practice, and fear

Spearman rank-order correlation tests explored that knowledge score was significantly correlated with attitude score (P < 0.01) although it was insignificantly correlated with other scores (practice and fear score). However, attitude score was positively correlated with both practice score (P < 0.001) and fear score (P < 0.001). Along with attitude score, practice score and fear score was also positively correlated with each other (P < 0.01) (Table 6). Multiple logistic regression analysis uncovered that the participants who had more accurate knowledge were

**Table 3. Fear related to covid-19 among respondents (N = 382) (distribution of responses).**

| Fear Questions | Strongly Disagree | Disagree | Neutral | Agree | Strongly Agree |
|---|---|---|---|---|---|
| | n (%) | n (%) | n (%) | n (%) | n (%) |
| I am most afraid of coronavirus-19. | 43 (11.3) | 96 (25.1) | 124 (32.5) | 90 (23.6) | 29 (7.6) |
| It makes me uncomfortable to think about coronavirus-19. | 35 (9.2) | 118 (30.9) | 93 (24.3) | 109 (28.5) | 27 (7.1) |
| My hands become clammy when I think about coronavirus-19. | 119 (31.2) | 216 (56.5) | 29 (7.6) | 15 (3.9) | 3 (0.8) |
| I am afraid of losing my life because of coronavirus-19. | 47 (12.3) | 134 (35.1) | 100 (26.2) | 81 (21.2) | 20 (5.2) |
| When watching news and stories about coronavirus-19 on social media, I become nervous or anxious. | 45 (11.8) | 117 (30.6) | 103 (27.0) | 86 (22.5) | 31 (8.1) |
| I cannot sleep because I'm worried about getting coronavirus-19. | 132 (34.6) | 204 (53.4) | 32 (8.4) | 13 (3.4) | 1 (0.3) |
| My heart races or palpitates when I think about getting coronavirus-19. | 127 (33.2) | 180 (47.1) | 49 (12.8) | 24 (6.3) | 2 (0.5) |

n- Number of respondents, %- Percentage of respondents, N- Total number of respondents

2.34 times more likely to have a more positive attitude (AOR = 2.34, 95% CI = 1.23–4.47, P < 0.01) and 2.17 times more likely to have very little fear (AOR = 2.17, 95% CI = 1.10–4.26, P < 0.05) than those who had less accurate knowledge. Participants who had a more positive attitude were 4 times more likely to have more frequent practice than those who had less positive attitude (AOR = 4.00, 95% CI = 2.44–6.56, P < 0.001). Moreover, the participants who had very less fear were 56% less likely to have more positive attitude (AOR = 0.44, 95% CI = 0.23–0.84, P < 0.01) and at the same time 53% less likely to have more frequent practice (AOR = 0.47, 95% CI = 0.26–0.84, P < 0.01) than those who had moderate to high fear (Table 5).

## Probable reasons explaining why students are detached from maintaining CPMs

When participants were asked whether they were maintaining safety rules for Covid-19, 27% (103) of participants responded 'Yes' to assure that they were maintaining safety rules completely while 73% (279) responded 'No' and mentioned their possible reasons as to why they were detached from maintaining safety rules properly. Of the participants who responded 'No', 56.3% mentioned that they could not maintain a social distance because they needed to go out for many reasons and needed to use public transports where social distance maintaining was not possible, while 44.8% of participants mentioned they needed to go to market for daily groceries or other things where they could not maintain social distance. 28.3% of participants thought of themselves as social butterflies and met with many friends while they did not maintain social distance. 22.2% of participants reported that they could not wash their hands regularly, while 32.0% mentioned that they washed their hands as recommended by WHO very few times a day. Further analysis explored that among participants who mentioned that they washed their hands regularly, 22.1% did not wash their hands as recommended by WHO, and among those who mentioned that they did not wash their hands regularly, 66.1% maintained rules as recommended by WHO when they washed their hands. In addition, 37.6% of participants did not use hand sanitizers when staying outside the home and 23.3% of participants had bad habits of touching eyes, nose, mouth, frequently. 21.1% of participants reported that

**Table 4. Association of socio-demographic characteristics with knowledge, attitude, fear, and practice score.**

| Socio-demographic characteristics (n) | Knowledge | | Attitude | | Fear | | Practice | |
|---|---|---|---|---|---|---|---|---|
| | Mean Rank | U/H test P-value | Mean Rank | U/H test P-value | Mean Rank | U/H test P-value | Mean Rank | U/H test P-value |
| **Sex**\* | | | | | | | | |
| Male (285) | 195.32 | 0.225 | 186.58 | 0.130 | 176.37 | **<0.001** | 175.86 | **<0.001** |
| Female (97) | 180.26 | | 205.95 | | 235.96 | | 237.46 | |
| **Education**\* | | | | | | | | |
| HSC (104) | 183.24 | 0.349 | 200.50 | 0.323 | 204.98 | 0.144 | 206.07 | 0.112 |
| BSc/MSc/Above(278) | 194.59 | | 188.13 | | 186.46 | | 186.05 | |
| **Occupation**\* | | | | | | | | |
| Student and self-employed (152) | 197.62 | 0.357 | 177.89 | **0.047** | 181.38 | 0.144 | 180.61 | 0.115 |
| Student only (230) | 187.45 | | 200.49 | | 198.19 | | 198.70 | |
| **Family type**\* | | | | | | | | |
| Nuclear (293) | 188.11 | 0.225 | 187.76 | 0.224 | 188.37 | 0.313 | 194.41 | 0.347 |
| Extended (89) | 202.65 | | 203.80 | | 201.81 | | 181.93 | |
| **Income**\*\* | | | | | | | | |
| <20,000 BDT (222) **(a)** | 187.35 | 0.647 | 179.91 | **0.05** | 194.48 | 0.622 | 173.86 | **0.001** |
| 20,000–40,000 BDT (126) **(b)** | 196.30 | | 207.26 | | 183.97 | | 212.09 | |
| >40,000 BDT (34) € | 200.82 | | 208.76 | | 199.97 | | 230.37 | |
| **Location**\*\* | | | | | | | | |
| Village (142) **(d)** | 182.30 | 0.360 | 191.30 | 0.425 | 183.32 | 0.484 | 166.96 | **0.004** |
| Sub-district town (39) **(e)** | 210.58 | | 197.91 | | 182.69 | | 189.85 | |
| District town (97) **(f)** | 187.76 | | 203.31 | | 204.63 | | 217.15 | |
| Divisional district town (104) | 200.39 | | 178.35 | | 193.73 | | 201.70 | |

B–T—Bangladesh Taka (Currency symbol of Bangladesh taka), HSC = Higher Secondary School Certificate, BSc- Bachelor of Science, MSc- Masters of Science, p-Significance value

\*U = Mann-Whitney U

\*\*H = Kruskal-Wallis H

[**(a)** vs **(b)** and **(a)** vs **(c)** were significant at 0.005 and 0.015 level respectively in practice (Bonferroni corrected)**; (a)** vs **(b)** was significant at 0.024 level in attitude (without Bonferroni correction)**; (d)** €**(e)** was significant at 0.003 in practice (Bonferroni corrected)**; (d)** vs **(f)** was significant at 0.014 level in practice (without Bonferroni correction)]

they had not seen anyone from their near or distant relatives or neighbors getting infected by a coronavirus and that was why they did not give importance to maintaining safety rules of Covid-19. Fig 2 depicted all other reasons graphically with the frequency of responses for each reason and Table 7 tabulated all the supportive information.

## Discussion

The findings showed that participants' overall correct answer rate was 89.6%, indicating participants had appreciable knowledge on Covid-19. This outcome is consistent with previous studies conducted last year (2020) in Bangladesh (81%) [15], China (90%) [16], Malaysia (80.5%) [17], Saudi Arabia (82%) [18], Iran (90% & 85%) [19], and Nepal (60% to 99%) [20]. Moreover, 84.8% of participants had more accurate knowledge which is higher compared to that reported by some previous Bangladeshi investigations [4–8], indicating knowledge level has elevated among Bangladeshis. We found that knowledge scores do not significantly vary with sex, education, occupation, location, income, or family type, which is inconsistent with some studies [4–8, 16, 21].

**Table 5. Results of multiple logistic regression on factors associated with knowledge, attitude, practice, and fear.** (Adjusted model).

| Variables (D) | More | | More | | More | | Very little fear | |
| --- | --- | --- | --- | --- | --- | --- | --- | --- |
| | Accurate knowledge | | Positive attitude | | frequent practice | | | |
| | n (%) | AOR (95% CI) | n (%) | AOR (95% CI) | n (%) | AOR (95% CI) | n (%) | AOR (95% CI) |
| **Sex** | | | | | | | | |
| Male (285) | 243 (85.3) | 1.14 (0.56–2.32) | 182 (63.9) | 1.22 (0.67–2.20) | 140 (49.1) | **0.51* (0.29–0.89)** | 232 (81.4) | **1.89* (1.05–3.40)** |
| Female (97) | 81 (83.5) | ref | 70 (72.2) | ref | 69 (71.1) | ref | 65 (67.0) | ref |
| **Education** | | | | | | | | |
| HSC (104) | 91 (87.5) | ref | 74 (71.1) | ref | 62 (59.6) | ref | 79 (76.0) | ref |
| BSc/MSc/Above (278) | 233 (83.8) | 0.73 (0.36–1.49) | 178 (64.0) | 0.83 (0.47–1.45) | 147 (52.9) | 0.94 (0.55–1.62) | 218 (78.4) | 0.96 (0.53–1.74) |
| **Occupation** | | | | | | | | |
| Student and self-employed (152) | 130 (85.5) | ref | 93 (61.2) | ref | 74 (48.7) | ref | 124 (81.6) | ref |
| Student only (230) | 194 (84.3) | 0.90 (0.47–1.73) | 159 (69.1) | 1.09 (0.65–1.84) | 135 (58.7) | 1.20 (0.73–1.99) | 173 (75.2) | 0.87 (0.48–1.58) |
| **Family type** | | | | | | | | |
| Nuclear (293) | 246 (84.0) | ref | 189 (64.5) | ref | 163 (55.6) | ref | 230 (78.5) | ref |
| Extended (89) | 78 (87.6) | 1.45 (0.70–3.01) | 63 (70.8) | 1.28 (0.73–2.25) | 46 (51.7) | 0.86 (0.50–1.46) | 67 (75.3) | 0.73 (0.40–1.33) |
| **Income** | | | | | | | | |
| <20,000 BDT (222) | 184 (82.9) | 0.89 (0.27–3.0) | 135 (60.8) | **0.33* (0.11–0.94)** | 103 (46.4) | 0.63 (0.26–1.55) | 177 (79.7) | 0.53 (0.19–1.40) |
| 20,000–40,000 BDT (126) | 110 (87.3) | 1.09 (0.31–3.78) | 89 (70.6) | 0.41 (0.14–1.19) | 81 (64.3) | 1.06 (0.42–2.65) | 93 (73.8) | 0.50 (0.19–1.33) |
| >40,000 BDT (34) | 30 (88.2) | ref | 28 (82.4) | ref | 25 (73.5) | ref | 27 (79.4) | ref |
| **Location** | | | | | | | | |
| Village (142) | 113 (79.5) | ref | 92 (64.8) | ref | 63 (44.4) | ref | 115 (81.0) | ref |
| Sub-district town (39) | 37 (94.9) | **4.85* (1.07–22.05)** | 26 (66.7) | 0.844 (0.37–1.93) | 19 (48.7) | 0.98 (0.44–2.19) | 31 (79.5) | 0.92 (035–2.40) |
| District town (97) | 83 (85.6) | 1.46 (0.69–3.08) | 71 (73.2) | 0.99 (0.53–1.89) | 63 (64.9) | **2.01* (1.09–3.65)** | 72 (74.2) | 0.78 (0.39–1.54) |
| Divisional district town (104) | 91 (87.5) | 2.03 (0.93–4.40) | 63 (60.6) | **0.50* (0.27–0.93)** | 64 (61.5) | **2.04* (1.12–3.73)** | 79 (76.0) | 0.66 (0.33–1.31) |
| **Knowledge** | | | | | | | | |
| More accurate knowledge (324) | - | - | 222 (68.5) | **2.34** (1.23–4.47)** | 179 (55.2) | 0.94 (0.49–1.80) | 257 (79.3) | **2.17* (1.10–4.26)** |
| Less accurate knowledge (58) | - | - | 30 (51.7) | ref | 30 (51.7) | ref | 40 (69.0) | ref |
| **Attitude** | | | | | | | | |
| More positive attitude (252) | 222 (88.1) | **2.39** (1.26–4.54)** | - | - | 168 (66.7) | **4.00*** (2.44–6.56)** | 183 (72.6) | **0.43** (0.22–0.82)** |
| Less positive attitude (130) | 102 (78.5) | ref | - | - | 41 (31.5) | ref | 114 (87.7) | ref |
| **Practice** | | | | | | | | |
| More frequent practice (209) | 179 (85.6) | 0.98 (0.51–1.88) | 168 (80.4) | **4.01*** (2.44–6.55)** | - | - | 146 (69.9) | **0.46** (0.26–0.83)** |
| Less frequent practice (173) | 145 (83.8) | ref | 84 (48.6) | ref | - | - | 151 (87.3) | ref |
| **Fear** | | | | | | | | |
| High to moderate fear (85) | 67 (78.8) | ref | 69 (81.1) | ref | 63 (74.1) | ref | - | - |
| Very little fear (297) | 257 (86.5) | **2.25* (1.13–4.47)** | 183 (61.6) | **0.44** (0.23–0.84)** | 146 (49.2) | **0.47** (0.26–0.84)** | - | - |
| **Total** | 324 (84.8%) | | 252 (66.0%) | | 209 (54.7%) | | 297 (77.7%) | |

D = Number of respondents in each category of independent variables, n = Number of respondents in category of dependent variables, BSc- Bachelor of Science, MSc- Masters of Science, AOR = Adjusted Odd ratio, Reference category = ref

*P ≤ 0.05

**P ≤ 0.01

***P ≤ 0.001, High to moderate fear = equal or more than 50% fear score, Very less fear = less than 50% fear score.

Approximately 66% of participants had a more positive attitude toward Covid-19 prevention and control, which is commensurate with a study conducted in Bangladesh [6]. However, when compared to the other studies from Bangladesh [5, 7], it seems attitude level among students has alleviated to some extent. Compared to overseas investigations, our finding is similar to a study from Pakistan (65.4%) [10], while dissimilar with other studies from Saudi Arabia

**Table 6. Results of spearman's rank-order correlation among knowledge, attitude, practice, and fear score.**

|  | 1 | 2 | 3 | 4 |
|---|---|---|---|---|
| **1**. Knowledge | 1 |  |  |  |
| **2**. Attitude | 0.12[a] | 1 |  |  |
|  | 0.015[b] |  |  |  |
| **3**. Practice | 0.05[a] | 0.36[a] | 1 |  |
|  | 0.369[b] | < 0.001[b] |  |  |
| **4**. Fear | -0.05[a] | 0.24[a] | 0.15[a] | 1 |
|  | 0.280[b] | < 0.001[b] | 0.005[b] |  |

a = Correlation coefficient, b = Significance level (Two tailed)

(95%) [22], and China (73.8%) [12]. Our findings indicated that Bangladeshi students might have lost their confidence toward controlling the rapid escalation of coronavirus as only 35.3% of participants were confident in believing that Bangladesh could win the battle against Covid-19 and concomitantly, only 44.2% of participants were confident that Covid-19 would finally be controlled. Rabbani et al. [8] reported that 55.3% of their participants were confident about that Bangladesh would win the battle against Covid-19 which is also consistent with Kundu et al. [23], and 68.5% were confident that Covid-19 would finally be controlled which is somewhat lower compared to Banik et al. [7]. On the other hand, Malaysian, Chinese, Saudi Arabian, Indonesian, Nepali studies obtained, respectively, 83%, 91%, 94%, 94%, and 71.5% positive responses regarding whether Covid-19 would successfully be controlled, and 96%, 97%, 97%, 95.5%, and 80% positive response regarding whether their country would be able to win the battle against Covid-19 [16–18, 21, 24]. On the whole, it is obtrusive that attitude level among students subsided noticeably. The reason behind obtaining such ominous outcomes might be the long duration of uncertainties which appeared due to emergence of new variants of SARS-COV-2 with the function of time. Furthermore, our study explored that attitude levels significantly varied with occupation, and family income of the participants, not with other socio-demographic variables included in our scrutiny, which is somewhat dissimilar with several studies. [4, 6, 7].

On average, participants had a 35.8% fear score which is lower compared to Hossain et al. [25]. Besides, only 2.1% of participants had very high fear and 77.7% had very less fear. These

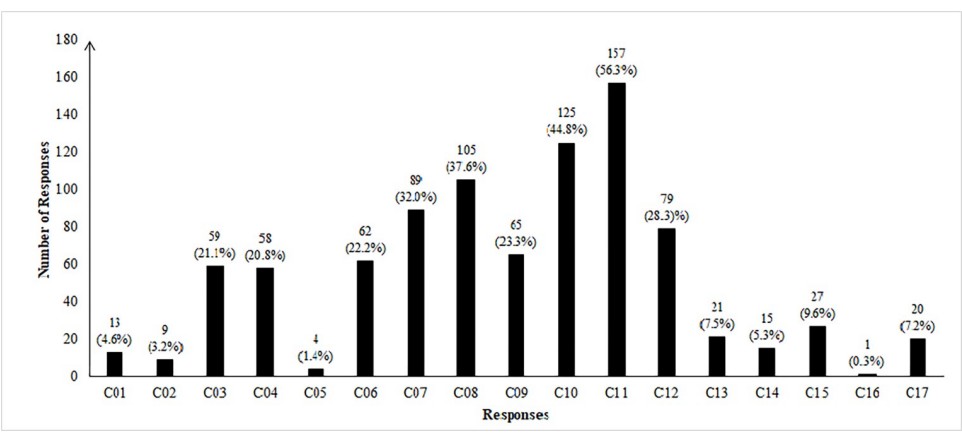

**Fig 2. Graphical representation of reasons behind failure to maintain safety guidelines of Covid-19.** Table 7 tabulated all the supportive information.

**Table 7. Possible reasons why people cannot maintain safety rules of Covid-19 completely.**

- Beliefs that I will not be infected by the coronavirus. **(C01)**

- Things related to corona seem to me as media-created rumors. **(C02)**

- No one having been identified as corona virus-positive yet from my near or distant relatives or neighbors makes maintaining safety rules less important to me. **(C03)**

- I am less panicked about coronavirus. **(C04)**

- I have no clear conception about safety rules for Covid-19. **(C05)**

- It is not always possible for me to wash hands coming from outsides. **(C06)**

- Very few times in a day, hands are washed up to the elbow for (20–30) min as recommended by WHO. **(C07)**

- I do not use hand sanitizer when staying outside. **(C08)**

- Trying but I can't give up the bad habit of touching eyes, face, nose, frequently. **(C09)**

- I am to go to the market often for daily groceries or other things where maintaining social distance is not possible. **(C10)**

- I am to go out for many reasons and use public transports where maintaining social distance is not possible. **(C11)**

- I am a social butterfly and hang out with friends very often while social distance is not maintained. **(C12)**

- I don't maintain social distance thinking about the benefit of maintaining social distance when people around me don't maintain social distance. **(C13)**

- I don't use medical masks because of having any confidence in a normal medical mask (5 BDT) as well as no money for buying expensive masks. **(C14)**

- When wearing a mask, I feel very uneasy and it's very hard to stay with. **(C15)**

- I don't use a mask because it looks ugly. **(C16)**

- I don't know any reason. **(C17)**

BDT-Bangladesh Taka (Currency symbol of Bangladeshi taka)

outcomes indicate that students are no longer panicked. Females had significantly high fear than males. Hossain et al. reported that the fear score significantly differs with sex, geography, and education, when female, urban-dwelling, and higher educated participants had high fear, which is partially similar to our findings [25].

On average, participants had 74.2% of total practice score and 55% of participants had more frequent practice of Covid-19 preventive measures (CPM), which is consistent with some earlier studies from Bangladesh [4, 6, 7, 23]. Thus, our findings indicate that the preventive practice level remains similar to what was in last year. However, compared to other studies from Bangladesh (24%) [8] and Pakistan (36.5%) [10], the findings indicate that participants had more frequent practice and seem to have lower frequent practice when compared to the studies from Saudi Arabia (81% & 82%) [11, 22] and South Korea (87.9%) [12]. Indeed, we found that practice frequency is greatly influenced by sex, family income, and location. However, the frequency of practice could also be influenced by other socio-demographic factors such as education, occupation, age, marital status, number of earning person in family, religion [5–8].

The finding of this study showed that knowledge regarding Covid-19 and CPM has an indirect effect on preventive practice mediated by attitude, as knowledge has a positive impact on attitude and concomitantly, a positive attitude increases preventive practice. Similar outcomes were reported in other studies [5, 8, 9, 22, 26, 27]. Moreover, our study uncovered that knowledge has a significant inverse effect on fear, indicating the more accurate the knowledge, the less the fear. This outcome corroborates with the findings from other simillar researches [15, 22, 25, 28]. Our study also revealed that like attitude, fear has also a positive impact on preventive practice, indicating the more the fear, the more the preventive practice. A similar outcome was reported by Hossain et al. and W/Mariam et al. [28, 29]. Ostensibly, this outcome could

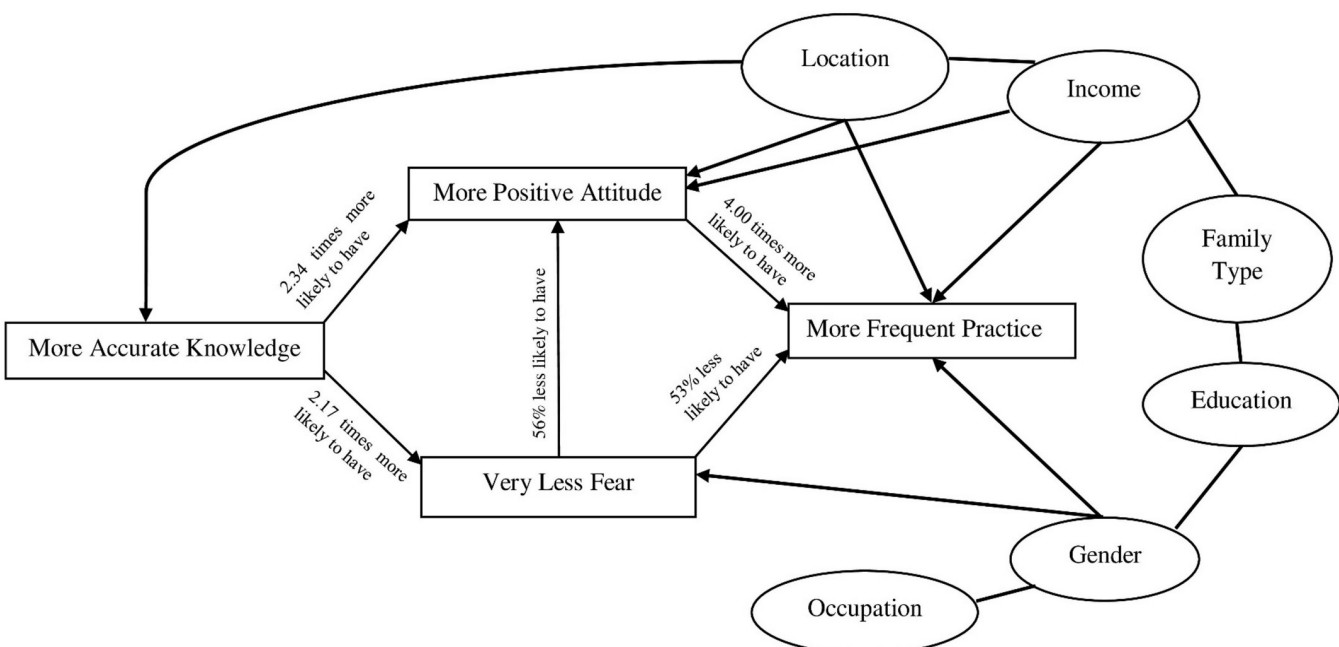

**Fig 3. Diagram showing the direct and indirect impact of knowledge, attitude, fear, and socio-demographic factors (Ellipse) on preventive practice.**

seem controversial but it actually represents the significant influence of fear on changing someone's behavior or attitude toward preventive practice. However, fear has shortcomings too. For instance, Covid-19 fear is significantly associated with anxiety and psychological distress, which could in turn trigger stress and mental illness [28]. Besides, there was an instance of suicidal death due to the fear of Covid-19 in Bangladesh [30]. Fig 3 depicted all the factors and their impact on preventive practice.

## Limitations, strength, and recommendation

Our study had several limitations. First, our study sample could not reflect general population of the country. Second, the study sample size was comparatively small. Third, our study followed a web-based cross-sectional study that excluded underprivileged students especially those who were not accessible to internet-based facilities. Furthermore, we had to conduct comparatively less sensitive statistical analysis as our data were not normally distributed. As to the strength of our study, to the best of our knowledge, this was one of the first studies that attempted to find out possible reasons why students were deflected from maintaining preventive measures. Future investigations should follow both community-based and web-based cross-sectional studies, collecting data from all sectors, all divisions, attempting to find out the most probable reasons behind failure to maintain all CPM in a more structured way.

## Conclusion

Our study attempted to bring up the present knowledge, attitude, practice, and fear level of Bangladeshi students. According to our study, students had appreciable knowledge and very little fear regarding Covid-19 but disappointedly had average attitude and practice toward Covid-19 prevention. Besides, students lacked confidence that Bangladesh could win the battle against Covid-19. Therefore, it is time for public health policymakers to be more focused to scale up measures to increase people's confidence regarding the ability of the health system of

Bangladesh toward the fight against Covid-19 or any future disease outbreaks. They should also strive to improve knowledge, attitude, and practice regarding personal protective measures against any infection. In doing so, they must be more dogmatic to develop rules and regulations, and more austere in implementing and maintaining those to combat any new onslaught of the deadly coronavirus or other viruses. Moreover, providing public health educations, arranging viable public health campaigns, promoting accurate information regarding Covid-19, counseling students as well as general people on preventive practices, must be continued till the end of the unprecedented condition.

## Supporting information

**S1 File. Questionnaire and coding information.**
(DOCX)

**S2 File. Covid-19 raw data.**
(XLSX)

## Acknowledgments

The authors acknowledge all the participants who participated in the study and all the volunteers who helped in data collection. The authors are also grateful to Prof. Dr. Md. Golam Sadik and Dr. Md. Abdur Rafi for their support in the overall improvement of the manuscript.

## Author Contributions

**Conceptualization:** Tahsin Ahmed Rupok, Sunandan Dey.

**Data curation:** Tahsin Ahmed Rupok, Sunandan Dey, Rashni Agarwala, Md. Nurnobi Islam, Bayezid Bostami.

**Formal analysis:** Tahsin Ahmed Rupok, Rashni Agarwala.

**Investigation:** Tahsin Ahmed Rupok, Sunandan Dey, Rashni Agarwala, Md. Nurnobi Islam.

**Methodology:** Tahsin Ahmed Rupok, Bayezid Bostami.

**Project administration:** Tahsin Ahmed Rupok, Sunandan Dey.

**Software:** Tahsin Ahmed Rupok.

**Supervision:** Tahsin Ahmed Rupok, Sunandan Dey.

**Writing – original draft:** Tahsin Ahmed Rupok.

**Writing – review & editing:** Tahsin Ahmed Rupok, Sunandan Dey, Rashni Agarwala, Md. Nurnobi Islam, Bayezid Bostami.

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
