## [Decision Letter · Decision Letter 0]

17 Dec 2021

PONE-D-21-23855Present knowledge, attitude, practice, and fear level of Bangladeshi people towards covid-19 after a year of the pandemic situation: a web-based cross-sectional study.PLOS ONE

Dear Dr. Rupok,

Thank you for submitting your manuscript to PLOS ONE. After careful consideration, we feel that it has merit but does not fully meet PLOS ONE’s publication criteria as it currently stands. Therefore, we invite you to submit a revised version of the manuscript that addresses the points raised during the review process.

The reviewers identified some major concern and those need to be fixed before arriving final decision.

We look forward to receiving your revised manuscript.

Kind regards,

Enamul Kabir

Academic Editor

PLOS ONE

Reviewers' comments:

Reviewer's Responses to Questions

**Comments to the Author**

1. Is the manuscript technically sound, and do the data support the conclusions?

Reviewer #1: Yes

Reviewer #2: Partly

2. Has the statistical analysis been performed appropriately and rigorously? 

Reviewer #1: Yes

Reviewer #2: Yes

3. Have the authors made all data underlying the findings in their manuscript fully available?

Reviewer #1: Yes

Reviewer #2: Yes

4. Is the manuscript presented in an intelligible fashion and written in standard English?

Reviewer #1: No

Reviewer #2: No

5. Review Comments to the Author

Reviewer #1: As the COVID-19 pandemic is going through a second and third peak in Bangladesh, people’s perception about COVID-19 can work as an important mediator of its transmission dynamics. Hence, the topic studied by the authors is a timely one. The authors have done an online survey of mainly students where the knowledge, attitude, practice, and fear level towards Covid-19 was assessed. However there are several issues that has to be addressed to make it a standard one.

1. As more than 90% participants was students, the title should replace the word ‘people’ with ‘student’. Because the sample in no way represent the people of Bangladesh. So the title should be -

“Present knowledge, attitude, practice, and fear level of Bangladeshi students towards Covid-19 after a year of the pandemic situation: a web-based cross-sectional study.”

2. The authors must exclude other participants and include only students in the analysis. Therefore a re-analysis of data after exclusion of others should be done.

3. The introduction should be re-written to properly address the background and rationale.

4. The cut-off point for categorization of knowledge, attitude, practice, and fear is not clear. For example, why 81% was selected as a cut-off point for practice? It should be made clear in the methods section.

5. Why location was categorized in this way? What is the reason behind taking ‘Dhaka’ as a different location?

6. Result and discussion should be modified in light of new analysis.

7. Several queries are given in the manuscript draft.

8. The manuscript English needs improvement.

Reviewer #2: Overall comments:

The topic is of interest but several papers on this issue have already been published, so it is uncertainty to what extent the article will contribute to add knowledge to this field. For example, please see the link below:

https://journals.plos.org/plosone/article?id=10.1371/journal.pone.0239254

Substantial English edits are required

Specific comments to the authors:

Title:

Better to omit the word ‘Present’ from the title. It is suggested to keep the title as ‘Knowledge, attitude, practice, and fear level of Bangladeshi people towards covid-19 after a year of the pandemic situation: a web-based cross-sectional study’

This study has been conducted mainly among the students (out of 402 samples, 382 are students) and does not necessarily represent Bangladeshi people as a whole, therefore, it is suggested to make changes in the title of the manuscript.

Abstract:

Age of the study participants ranged from 16-30 meaning that all were not adults, so the authors are requested to make correction in the abstract and report the percentage of the participants <18 years.

Please report the exact percentage, not like more than 90%.

Background:

Overall, the background section needs to be re-written. It does not well justify the study.

Please do not mention individual’s name that can be identified easily and give reference of the statement. if it is personal communication, follow the reference style for that as well (lines 59-65)

Please mention the exact year instead of ‘this year’ (line 76)

The authors should mention ‘The Government of Bangladesh (GoB)’ throughout the paper, instead of ‘Bangladesh’s government’

Please mention the exact year instead of ‘last year’ (line 82)

Please give reference for the statement (lines 81-86) and follow the reference style instructed in author guideline

I don’t see any figure or diagram in the document (missing!)

Methodology:

It is not clear how the participants were approached and selected.

It seems that the study was conducted among the Bangladeshi student. Please make it clear, make correction accordingly throughout the paper.

Please write ‘participants’ or ‘respondents’ instead of study ‘subjects’

Income is not sufficient to determine social status and classify as ‘lower class, middle class and high class’. It is suggested to keep it simple as ‘income’ and categorize the income using range.

Ethical consideration:

The authors should mention how they have taken assents from the participants of <18 years

Data management and analysis:

The author should mention why they performed different statistical test and what they found from each test (i.e., how was the distribution of data under the variable(s)?)

What was/were the outcome(s) of interest in regression model? what independent variables were taken into consideration? And level of significance considered during keeping the variables in final model.

The authors performed MLR; why didn’t they report adjusted OR?

Results:

Overall, the results are not well interpreted. The authors have been suggested organize, present and interpret the results in scientific manner. Examples are numerous:

Please mention the equivalent grades instead of ‘honors and masters’ (line 207)

It seems that the participants were the students, not from other occupations. The authors have been suggested to rewrite and interpret the result section accordingly.

‘Participants’ Overall correct answer rate was 89.6% [Mean Score (MS) ± Standard 218 Deviation (SD) = 10.75±1.20 out of 12] while the correct answer rate was 65% to 219 100%, indicating participants had appreciable knowledge on covid-19.’ Interpretation of the results is confusing. Please rephrase.

6. PLOS authors have the option to publish the peer review history of their article (what does this mean?). If published, this will include your full peer review and any attached files.

Reviewer #1: No

Reviewer #2: No

---

## [Author Response · Author response to Decision Letter 0]

22 Feb 2022

Here I am writing our responses to each points raised by the reviewers.

Responses to the queries of Reviewer #2:

Title: 

Better to omit the word ‘Present’ from the title. It is suggested to keep the title as ‘Knowledge, attitude, practice, and fear level of Bangladeshi people towards covid-19 after a year of the pandemic situation: a web-based cross-sectional study’

Author: I made changes as you suggested. Please vide the title page.

This study has been conducted mainly among the students (out of 402 samples, 382 are students) and does not necessarily represent Bangladeshi people as a whole, therefore, it is suggested to make changes in the title of the manuscript.

Author: I made changes as you suggested. Please vide the title page. I excluded all the participants who were not student. 

Abstract:

Age of the study participants ranged from 16-30 meaning that all were not adults, so the authors are requested to make correction in the abstract and report the percentage of the participants <18 years. 

Author: I made correction as you suggested. Please vide (line 38). I could not report exact percentage of the participants <18 as in our questionnaire, under the question “what is your age category?” there was options like below 16 or 16-30 or more than 30 years old. 

Please report the exact percentage, not like more than 90%.

Author: I made correction as you suggested. Please vide (line 38).

Background:

Overall, the background section needs to be re-written. It does not well justify the study.

Please do not mention individual’s name that can be identified easily and give reference of the statement. If it is personal communication, follow the reference style for that as well (lines 59-65)

Author: I made correction as you suggested. (Lines 60-65)

Please mention the exact year instead of ‘this year’ (Line 76)

Author: I made change as you suggested. (Line 71)

The authors should mention ‘The Government of Bangladesh (GoB)’ throughout the paper, instead of ‘Bangladesh’s government’

Author: I made change as you suggested (line 73)

Please mention the exact year instead of ‘last year’ (line 82)

Author: I made correction as you suggested (Line 77)

Please give reference for the statement (lines 81-86) and follow the reference style instructed in author guideline

Author: I did give reference. (Line 80)

I don’t see any figure or diagram in the document (missing!)

Author: Figures or diagrams were uploaded separately with name like “Fig 1, or Fig 2”

Methodology:

It is not clear how the participants were approached and selected.

Author: I reported how the participants were approached and selected. (Lines 122-126)

It seems that the study was conducted among the Bangladeshi student. Please make it clear, make correction accordingly throughout the paper.

Author: I did perform statistical analysis on the participants as you suggested and made correction throughout the paper accordingly.

Please write ‘participants’ or ‘respondents’ instead of study ‘subjects’

Author: I made correction as you said (line 127).

Income is not sufficient to determine social status and classify as ‘lower class, middle class and high class’. It is suggested to keep it simple as ‘income’ and categorize the income using range.

Author: I made correction as you suggested (136)

Ethical consideration:

The authors should mention how they have taken assents from the participants of <18 years

Author: Assents from the participants of <18 years were taken thorough online approaches as were taken from other participants. This study included only those who had online accesses. 

Data management and analysis:

The author should mention why they performed different statistical test and what they found from each test (i.e., how was the distribution of data under the variable(s)?)

Author: I mentioned the purposes of the different statistical tests (line 189-196), reporting the outcomes of those tests in result section. I mentioned the distribution of data under the variables (line 196-203)

What was/were the outcome(s) of interest in regression model? what independent variables were taken into consideration? And level of significance considered during keeping the variables in final model.

Author: I reported the outcomes of interest and independent variables taken in regression model (line 193—96). I mentioned level of significance for all the analysis in together (line 203).

The authors performed MLR; why didn’t they report adjusted OR?

Author: I made correction throughout the paper as you suggested.

Results:

Overall, the results are not well interpreted. The authors have been suggested organize, present and interpret the results in scientific manner. Examples are numerous:

Please mention the equivalent grades instead of ‘honors and masters’ (line 207)

Author: I made correction throughout the paper as you suggested (line 209).

It seems that the participants were the students, not from other occupations. The authors have been suggested to rewrite and interpret the result section accordingly. 

Author: I excluded participants from occupations other than student, performed analysis, and interpreted the result section accordingly.

‘Participants’ Overall correct answer rate was 89.6% [Mean Score (MS) ± Standard 218 Deviation (SD) = 10.75±1.20 out of 12] while the correct answer rate was 65% to 219 100%, indicating participants had appreciable knowledge on covid-19.’ Interpretation of the results is confusing. Please rephrase.

Author: I made correction as you suggested (line 220-222).

Responses to some queries of Reviewer #1:

Majority of the queries and suggestions were similar with Reviewer #2. Here I am mentioning those which were somewhat exceptional.

The cut-off point for categorization of knowledge, attitude, practice, and fear is not clear. For example, why 81% was selected as a cut-off point for practice? It should be made clear in the methods section.

Author: I made correction and mentioned in methods section (line 155-167).

Why location was categorized in this way? What is the reason behind taking ‘Dhaka’ as a different location?

Author: Although Dhaka is the divisional district like other divisions but initially Dhaka was taken as a different location because it is the capital of the Bangladesh, majority of people lives there so infection rate and death rate were exceptionally large than other divisions and districts. That is why Dhaka was so taken. However, as number of participants from this location was very small, therefore, during analysis participants from Dhaka were considered as participants from divisional district town and different tests were conducted accordingly.

Moreover, I made correction what you marked in manuscript draft.

---

## [Decision Letter · Decision Letter 1]

11 Mar 2022

PONE-D-21-23855R1Knowledge, attitude, practice, and fear level of Bangladeshi students towards covid-19 after a year of the pandemic situation: a web-based cross-sectional study.PLOS ONE

Dear Dr. Rupok,

Thank you for submitting your manuscript to PLOS ONE. After careful consideration, we feel that it has merit but does not fully meet PLOS ONE’s publication criteria as it currently stands. Therefore, we invite you to submit a revised version of the manuscript that addresses the points raised during the review process.

We look forward to receiving your revised manuscript.

Kind regards,

Enamul Kabir

Academic Editor

PLOS ONE

Journal Requirements:

Reviewers' comments:

Reviewer's Responses to Questions

**Comments to the Author**

1. If the authors have adequately addressed your comments raised in a previous round of review and you feel that this manuscript is now acceptable for publication, you may indicate that here to bypass the “Comments to the Author” section, enter your conflict of interest statement in the “Confidential to Editor” section, and submit your "Accept" recommendation.

Reviewer #1: All comments have been addressed

Reviewer #2: (No Response)

2. Is the manuscript technically sound, and do the data support the conclusions?

Reviewer #1: Partly

Reviewer #2: Yes

3. Has the statistical analysis been performed appropriately and rigorously? 

Reviewer #1: Yes

Reviewer #2: Yes

4. Have the authors made all data underlying the findings in their manuscript fully available?

Reviewer #1: Yes

Reviewer #2: Yes

5. Is the manuscript presented in an intelligible fashion and written in standard English?

Reviewer #1: No

Reviewer #2: No

6. Review Comments to the Author

Reviewer #1: This was a well conducted study and the results were presented and interpreted in an appropriate manner. You should do some additional minor corrections. All the comments are provided in the revised manuscript draft. However, the authors need to extensively review and edit grammars and writing. Please use help from a native English speaker, or from someone who is an expert user of English.

Reviewer #2: Comments to the author:

Substantial language edits are still required.

Research ethics: It is not clear how they have taken assents from the participants of under-18. Guardians' consents are also required in this case. It is not clear how they have reached out the guardian of under-18 students while conducting a web-based survey. This needs to be explained in the manuscript.

The author mentioned that the study was approved by the ethical review committee of

Rajshahi University. The authors are suggested to provide the approval letter as a supplementary file for strengthening the credibility of ethical issue of the study.

7. PLOS authors have the option to publish the peer review history of their article (what does this mean?). If published, this will include your full peer review and any attached files.

Reviewer #1: No

Reviewer #2: No

---

## [Author Response · Author response to Decision Letter 1]

14 Mar 2022

Here I am writing our responses to each points raised by the reviewers.

Responses to Reviewer #1

 This was a well conducted study and the results were presented and interpreted in an appropriate manner. You should do some additional minor corrections. All the comments are provided in the revised manuscript draft. However, the authors need to extensively review and edit grammars and writing. Please use help from a native English speaker, or from someone who is an expert user of English.

Author: Thanks for your valuable comments. We have corrected all the marked issues as you suggested (Please vide the Revised manuscript with track changes). We (all authors) thoroughly revised the manuscript again and also taken help from a public health researcher expert in this field as well as expert user of English. If further correction needed, please inform us we will correct according to your suggestion.

Responses to Reviewer #2 

Substantial language edits are still required.

Research ethics: It is not clear how they have taken assents from the participants of under-18. Guardians' consents are also required in this case. It is not clear how they have reached out the guardian of under-18 students while conducting a web-based survey. This needs to be explained in the manuscript.

The author mentioned that the study was approved by the ethical review committee of

Rajshahi University. The authors are suggested to provide the approval letter as a supplementary file for strengthening the credibility of ethical issue of the study.

Author: Thanks for your valuable suggestion. I have cleared how we had reached out the guardian of under 18 year old participants and other ethical issues in the manuscript [Please vide line (123-124) & (199-204)].

---

## [Decision Letter · Decision Letter 2]

16 May 2022

PONE-D-21-23855R2Knowledge, attitude, practice, and fear level of Bangladeshi students towards Covid-19 after a year of the pandemic situation: a web-based cross-sectional study.PLOS ONE

Dear Dr. Rupok,

Thank you for submitting your manuscript to PLOS ONE. After careful consideration, we feel that it has merit but does not fully meet PLOS ONE’s publication criteria as it currently stands. Therefore, we invite you to submit a revised version of the manuscript that addresses the points raised during the review process.

We look forward to receiving your revised manuscript.

Kind regards,

Enamul Kabir

Academic Editor

PLOS ONE

Journal Requirements:

Reviewers' comments:

Reviewer's Responses to Questions

**Comments to the Author**

1. If the authors have adequately addressed your comments raised in a previous round of review and you feel that this manuscript is now acceptable for publication, you may indicate that here to bypass the “Comments to the Author” section, enter your conflict of interest statement in the “Confidential to Editor” section, and submit your "Accept" recommendation.

Reviewer #1: All comments have been addressed

Reviewer #2: (No Response)

2. Is the manuscript technically sound, and do the data support the conclusions?

Reviewer #1: Yes

Reviewer #2: Yes

3. Has the statistical analysis been performed appropriately and rigorously? 

Reviewer #1: Yes

Reviewer #2: Yes

4. Have the authors made all data underlying the findings in their manuscript fully available?

Reviewer #1: Yes

Reviewer #2: Yes

5. Is the manuscript presented in an intelligible fashion and written in standard English?

Reviewer #1: Yes

Reviewer #2: No

6. Review Comments to the Author

Reviewer #1: After the adjustments of the previous reviews the manuscript has now come to a shape for publication. I think, it can now be accepted. However, it could have been better if some parts of the manuscript could be re-written in a more engaging language.

Reviewer #2: The authors are suggested to provide the approval letter as a supplementary file for strengthening the credibility of ethical issue of the study.

7. PLOS authors have the option to publish the peer review history of their article (what does this mean?). If published, this will include your full peer review and any attached files.

Reviewer #1: No

Reviewer #2: No

---

## [Author Response · Author response to Decision Letter 2]

27 May 2022

Dear editor, 

Here I am writing our responses to each points raised by the reviewers.

Responses to the queries of Reviewer #1: 

Thanks for your valuable comments. According to your suggestion, all the authors critically revised the manuscript again. Besides, we requested another expert (name mentioned on the Acknowledgment section) to revise our manuscript. Hopefully, his contribution would add significant value to this manuscript. (All the changes have been marked on the “Revised Manuscript with Track Changes”)

Responses to the queries of Reviewer #2:

Thanks for your valuable suggestion. I provided the approval letter (scanned copy of the original letter) as a supplementary file for strengthening the credibility of ethical issue of the study.

---

## [Decision Letter · Decision Letter 3]

14 Jul 2022

PONE-D-21-23855R3Knowledge, attitude, practice, and fear level of Bangladeshi students towards Covid-19 after a year of the pandemic situation: a web-based cross-sectional study.PLOS ONE

Dear Dr. Rupok,

Thank you for submitting your manuscript to PLOS ONE. After careful consideration, we feel that it has merit but does not fully meet PLOS ONE’s publication criteria as it currently stands. Therefore, we invite you to submit a revised version of the manuscript that addresses the points raised during the review process.

We look forward to receiving your revised manuscript.

Kind regards,

Enamul Kabir

Academic Editor

PLOS ONE

Journal Requirements:

Reviewers' comments:

Reviewer's Responses to Questions

**Comments to the Author**

1. If the authors have adequately addressed your comments raised in a previous round of review and you feel that this manuscript is now acceptable for publication, you may indicate that here to bypass the “Comments to the Author” section, enter your conflict of interest statement in the “Confidential to Editor” section, and submit your "Accept" recommendation.

Reviewer #1: All comments have been addressed

Reviewer #2: (No Response)

2. Is the manuscript technically sound, and do the data support the conclusions?

Reviewer #1: Yes

Reviewer #2: Yes

3. Has the statistical analysis been performed appropriately and rigorously? 

Reviewer #1: Yes

Reviewer #2: Yes

4. Have the authors made all data underlying the findings in their manuscript fully available?

Reviewer #1: Yes

Reviewer #2: Yes

5. Is the manuscript presented in an intelligible fashion and written in standard English?

Reviewer #1: No

Reviewer #2: No

6. Review Comments to the Author

Reviewer #1: Despite repeated request the authors failed to improve the manuscript's language and grammar. However, as the study have some interesting findings, I've extensively read and put my comments and corrections in the manuscript. Please incorporate them to improve the style.

Reviewer #2: Editing in English is still required.

It is suggested that the authors seek assistance from someone who is proficient in written English.

7. PLOS authors have the option to publish the peer review history of their article (what does this mean?). If published, this will include your full peer review and any attached files.

Reviewer #1: No

Reviewer #2: No

---

## [Author Response · Author response to Decision Letter 3]

20 Jul 2022

Reviewer #1: Despite repeated request the authors failed to improve the manuscript’s language and grammar. However, as the study have some interesting findings, I’ve extensively read and put my comments and corrections in the manuscript. Please incorporate them to improve the style.

Author: I am expressing my sincere gratitude for your valuable comments and corrections. I have incorporated all the corrections you had made to the manuscript. Please see the “Revised Manuscript with Track Changes”. Please let me know, if further correction is needed.

Reviewer #2: Editing in English is still required. 

It is suggested that the authors seek assistance from someone who is proficient in written English.

Author: Thanks for your suggestion. Previously, I had taken assistance from two person who are proficient in written English but probably they could not perceive the demand of this journal. This time, I took assistance of them again and changed some part of the manuscript according to their suggestions and the suggestions made by “Reviewer #1”. Hopefully, this revised manuscript will meet the specifications of the journal. If it fails again, please let me know.

---

## [Decision Letter · Decision Letter 4]

11 Oct 2022

PONE-D-21-23855R4Knowledge, attitude, practice, and fear level of Bangladeshi students toward Covid-19 after a year of the pandemic situation: a web-based cross-sectional study.PLOS ONE

Dear Dr. %Rupok%,

Thank you for submitting your manuscript to PLOS ONE. After careful consideration, we feel that it has merit but does not fully meet PLOS ONE’s publication criteria as it currently stands. Therefore, we invite you to submit a revised version of the manuscript that addresses the points raised during the review process.

We look forward to receiving your revised manuscript.

Kind regards,

Nadim Sharif, M.Sc.

Academic Editor

PLOS ONE

Journal Requirements:

Additional Editor Comments :

The English language of the article should be revised accordingly.

PLOS ONE formatting of the article should be precisely followed.

Reviewers' comments:

Reviewer's Responses to Questions

**Comments to the Author**

1. If the authors have adequately addressed your comments raised in a previous round of review and you feel that this manuscript is now acceptable for publication, you may indicate that here to bypass the “Comments to the Author” section, enter your conflict of interest statement in the “Confidential to Editor” section, and submit your "Accept" recommendation.

Reviewer #1: All comments have been addressed

Reviewer #2: (No Response)

2. Is the manuscript technically sound, and do the data support the conclusions?

Reviewer #1: Yes

Reviewer #2: Yes

3. Has the statistical analysis been performed appropriately and rigorously? 

Reviewer #1: Yes

Reviewer #2: Yes

4. Have the authors made all data underlying the findings in their manuscript fully available?

Reviewer #1: Yes

Reviewer #2: Yes

5. Is the manuscript presented in an intelligible fashion and written in standard English?

Reviewer #1: No

Reviewer #2: No

6. Review Comments to the Author

Reviewer #1: Thanks to the authors that they incorporated all of the comments in the previous review. However, the manuscript still needs improvement in its description and presentation (i.e., English). I suggest the authors to take help from a native English speaker or a professional English editing service.

Reviewer #2: The language in this revised manuscript is still unclear, although the reviewers have repeatedly suggested the authors edit the language. Examples are numerous:

Lines (65-70): 'If this Delta variant starts to escalate vehemently in Bangladesh as it did in India and Nepal, people stay lackadaisical in maintaining public health hygiene, and the government continues to develop an ill-conceived plan of action and fails to ensure proper implementation of preventive measures, then, the number of cases detected per day could rise to an uncontrollable digit, which can throw the situation into turmoil (4).'

Lines (81-85) 'A review of some previous studies conducted in Bangladesh reflected the truth that in last year (2020) Bangladeshi people were comparatively less knowledgeable, had a less positive attitude toward Covid-19, and as a consequence had less frequent preventive practice, which was consistent with a study of Pakistan but inconsistent with China and Saudi Arabia (7-14).'

Lines (130-134): ' A structured questionnaire was prepared in a Google Form and a link was generated, shared with all authors and other volunteers who were instructed properly as to how well they can use this form to recruit data with adequate consent from the participants and from the guardians of the participants aged under 18 years old.

Lines (263-266): 'Mann-Whitney and Kruskal-Wallis tests showed that knowledge scores do not vary with any socio-demographic variables included in our study, indicating all the groups of a variable

had quite similar knowledge on Covid-19 disease (Table 4).

7. PLOS authors have the option to publish the peer review history of their article (what does this mean?). If published, this will include your full peer review and any attached files.

Reviewer #1: No

Reviewer #2: No

---

## [Author Response · Author response to Decision Letter 4]

24 Dec 2022

Response to Reviewers:

Reviewer #1: Thanks to the authors that they incorporated all of the comments in the previous review. However, the manuscript still needs improvement in its description and presentation (i.e., English). I suggest the authors to take help from a native English speaker or a professional English editing service.

Author: Thanks for your suggestion. To speak frankly, we are very new researchers and this is our very first manuscript that has come at this stage. We had no known native English speakers as well as no fund for taking any professional English editing service. We are mainly taking help from our teachers who are Phd holders and established researchers. Therefore, we request you kindly to consider our situation. If we could publish this manuscript, that would give us more confidence for future researches. 

Reviewer #2: The language in this revised manuscript is still unclear, although the reviewers have repeatedly suggested the authors edit the language. Examples are numerous:

Author: According to your suggestion, we reviewed our manuscript by another expert. According to his suggestion, we made changes many sentences to remove ambiguity. Please, see the “Revised Manuscript with Track Changes”.

Lines (65-70): 'If this Delta variant starts to escalate vehemently in Bangladesh as it did in India and Nepal, people stay lackadaisical in maintaining public health hygiene, and the government continues to develop an ill-conceived plan of action and fails to ensure proper implementation of preventive measures, then, the number of cases detected per day could rise to an uncontrollable digit, which can throw the situation into turmoil (4).'

Author: I have made changes. Please, see the Lines (73-76).

Lines (81-85) 'A review of some previous studies conducted in Bangladesh reflected the truth that in last year (2020) Bangladeshi people were comparatively less knowledgeable, had a less positive attitude toward Covid-19, and as a consequence had less frequent preventive practice, which was consistent with a study of Pakistan but inconsistent with China and Saudi Arabia (7-14).'

Author: I have made changes. Please, see the lines (86-88).

Lines (130-134): ' A structured questionnaire was prepared in a Google Form and a link was generated, shared with all authors and other volunteers who were instructed properly as to how well they can use this form to recruit data with adequate consent from the participants and from the guardians of the participants aged under 18 years old.

Author: I have made changes. Please, see the lines (128-130).

Lines (263-266): 'Mann-Whitney and Kruskal-Wallis tests showed that knowledge scores do not vary with any socio-demographic variables included in our study, indicating all the groups of a variable

had quite similar knowledge on Covid-19 disease (Table 4).

Author: I have made changes. Please, see the lines (258-260).

---

## [Editor Report · Decision Letter 5]

4 Jan 2023

PONE-D-21-23855R5Knowledge, attitude, practice, and fear level of Bangladeshi students toward Covid-19 after a year of the pandemic situation: a web-based cross-sectional study.PLOS ONE

Dear Dr. Rupok,

Thank you for submitting your manuscript to PLOS ONE. After careful consideration, we feel that it has merit but does not fully meet PLOS ONE’s publication criteria as it currently stands. Therefore, we invite you to submit a revised version of the manuscript that addresses the points raised during the review process.

We look forward to receiving your revised manuscript.

Kind regards,

Nadim Sharif, M.Sc.

Academic Editor

PLOS ONE

Journal Requirements:

Additional Editor Comments:

The discussion section should be revised accordingly.

The figure citation from discussion section should be removed.

Sub headings from discussion should be removed. PLOS ONE manuscript writing guidelines should be followed.

Can be found here, https://journals.plos.org/plosone/s/submission-guidelines

The authors should follow the reference style according to PLOS ONE guidelines.
---

## [Author Response · Author response to Decision Letter 5]

6 Jan 2023

Additional Editor Comments:

The discussion section should be revised accordingly.

The figure citation from discussion section should be removed.

Author: Due to some confusion, I have not removed figure citation from discussion section. I am not clear whether I should completely remove Fig. 3 and figure citation from the manuscript or replace in other section. I think, there should have a figure for clear conception. However, I will change according to your decision.

Sub headings from discussion should be removed. PLOS ONE manuscript writing guidelines should be followed. Can be found here, https://journals.plos.org/plosone/s/submission-guidelines

Author: According to suggestion, I removed all sub headings from discussion.

The authors should follow the reference style according to PLOS ONE guidelines.

Author: Reference style is ‘Vancouver’ which is in accordance with Plos One guidelines.

---

## [Editor Report · Decision Letter 6]

14 Feb 2023

Knowledge, attitude, practice, and fear level of Bangladeshi students toward Covid-19 after a year of the pandemic situation: a web-based cross-sectional study.

PONE-D-21-23855R6

Dear Dr. Rupok,

We’re pleased to inform you that your manuscript has been judged scientifically suitable for publication and will be formally accepted for publication once it meets all outstanding technical requirements.

Kind regards,

Nadim Sharif, M.Sc.

Academic Editor

PLOS ONE

Additional Editor Comments (optional):

Remove the Figure 3 from discussion and replace the figure 3 in result section.
---

## [Editor Report · Acceptance letter]

17 Feb 2023

PONE-D-21-23855R6 

Knowledge, attitude, practice, and fear level of Bangladeshi students toward Covid-19 after a year of the pandemic situation: a web-based cross-sectional study. 

Dear Dr. Rupok:

I'm pleased to inform you that your manuscript has been deemed suitable for publication in PLOS ONE. Congratulations! Your manuscript is now with our production department. 

Kind regards, 

on behalf of

Dr. Nadim Sharif 

Academic Editor

PLOS ONE